# Risk Decision Making and Executive Function among Adolescents and Young Adults

**DOI:** 10.3390/bs13020142

**Published:** 2023-02-08

**Authors:** Francisco Marquez-Ramos, David Alarcon, Josue G. Amian, Cristina Fernandez-Portero, Maria J. Arenilla-Villalba, Jose Sanchez-Medina

**Affiliations:** Department of Social Anthropology, Psychology and Public Health, Pablo de Olavide University, Carretera Utrera Km 1, Building 11, 41013 Seville, Spain

**Keywords:** dual system model, executive function, decision-making, Iowa Gambling Task, risk behavior, adolescent

## Abstract

The dual theory establishes that the decision-making process relies on two different systems, the affective system and the executive function (EF), developed during adolescence. This study analyzes the relationship between the decision-making and EF processes in a group of early adolescents (mean age = 12.51 years, SD = 0.61), where more affective impulse processes are developed, and in young adults (mean age = 19.38 years, SD = 1.97), where cognitive control processes have already matured. For this purpose, 140 participants in Spain completed the Iowa Gambling Task (IGT) to measure their risky decisions and the Wisconsin Card Sorting Task (WCST) to measure their EF. Performance on the IGT improves over blocks; however, adolescents received lower mean scores than young adults. However, controlling for age, perseverative errors were negatively associated with the mean net score on the risky blocks of IGT; thus, those who committed more perseverative errors in the WCST were more likely to take cards from the disadvantageous decks on the last blocks of the IGT. The current study shows that adolescents and adults solve ambiguous decisions by trial and error; however, adolescents are more likely to make risky decisions without attending to the long-term consequences. Following the dual theory hypothesis, the maturation of EF with age partly accounts for this difference in risky decision-making between adolescents and adults.

## 1. Introduction

Adolescence brings risky behaviors, such as experimentation with alcohol, tobacco and other drugs [1]. Similarly, a growing number of new types of risk behaviors have been observed among adolescents, such as gambling, careless sexual behavior and cyberbullying [2]. Some risk behaviors that occur in adolescence, such as drug use or certain eating disorders, are largely influenced by social and media pressures [3,4]. However, it is important to know which cognitive factors, apart from the sociocultural ones, favor the adolescent trend to make risky decisions. Adolescence is a stage of increased vulnerability, in which there is an accelerated transition of physical, biological, intellectual and social changes, which raises the probability of risky behaviors, such as the use of illegal substances. The Survey on Drug Use in Secondary Education in Spain [5] has reported alarming rates of alcohol use (53.6%), tobacco use (23.9%) and marijuana use (23.9%) in Spanish adolescents aged 14 to 18 years in the last 30 days. Espada et al. [6] observed in a sample of Spanish adolescents aged 15–18 years an increase in high-risk sexual practices between 2006 and 2012. Similar results have recently been observed in relation to risky behaviors on the road among young people in Spain [7].

Usually, risky behaviors are characterized as being rewarded in the short term but harmful in the long term [8]. Hence, risky behavior can be defined as those decisions where immediate satisfaction is preferred but without considering the long-term negative consequences [9]. Bechara et al. [10] demonstrate that the control of risky behavior is located in the ventromedial prefrontal cortex. According to these authors, patients with ventromedial prefrontal cortex lesions, in situations of uncertainty, often make decisions based on the immediate consequences without considering the long-term implications. Bechara et al. [10] proposed the somatic marker hypothesis to explain this insensitivity to future consequences in decision-making, which they called “myopia toward the future”. Subsequent studies corroborated these findings in patients with brain damage in the ventromedial prefrontal cortex [11]. In addition to patients with brain damage, this insensitivity to the future has been found in subjects with addictions. Thus, Johnson et al. [12] found that adolescents who consume alcoholic beverages also have this “myopia”, which prevents them from making advantageous decisions in the future due to a hypersensitivity to rewards. This concept was also used by Mogedas and Alameda [13], who, after studying the decision-making process in people with some type of addiction, found that 75% of them did not discern beyond the short-term reward, being unable to observe that long-term rewards were much better.

Several studies have shown that throughout adolescence, as the individual matures his or her cognitive control, an improvement in risky decision-making is observed [14,15,16]. From a neurocognitive perspective, the maturation of the prefrontal cortex during adolescence underlies the progress observed in executive function (EF) [17,18,19]. Then, the decrease in risky decisions, with the arrival of adulthood, may be associated with the increasing development of EF during adolescence. This study analyzed whether the differences in the decision-making of risks between young adults and adolescents are associated with improvements in their EF performance.

### 1.1. The Dual System Model

The dual system model could account for the relationship between the development of EF and the differences in risky decision-making among young adults and adolescents. According to this model, the teenager is in a process of maturational development of two neurological systems with different maturation rates, but these systems are competing when the teenager is making decisions [9,20,21]. The first of the neurological systems is responsible for reward processing and reward sensitivity, which is based on the subcortical brain structures, such as the striatum and amygdala, as well as the medial orbital and prefrontal cortices [22]. Reward sensitivity matures rapidly at the onset of puberty and triggers automatic emotional responses to certain contextual stimuli, leading to the selection of risky choices [9,14,23,24].

During adolescence, the neurological system in charge of the EF processes, which are related to the control, regulation and inhibition of these affective impulses, also matures. These EF processes are linked to the maturation of the neurological cognitive-control system, whose neural basis is in the lateral prefrontal, lateral parietal and anterior cingulate cortices [20,25,26]. However, the rate of development of the EF system is slower than that of the affective system, maturing throughout adolescence and even into early adulthood. This produces an imbalance in risky decision-making situations where, among young adolescents, a response is triggered in the affective system that is not sufficiently counterbalanced by the cognitive control system since the latter has not yet matured sufficiently enough to block the affective impulse [27,28].

According to the dual system model, the improvements seen in adolescents in decision-making upon reaching adulthood are due, in part, to the maturation of the interrelation systems between the affective or emotional system and the EF [9,29].

Following the dual model, the maturation of the PFC underlying the progressive development of the EF in adolescence could play an essential role in adolescents, increasing their ability to attend to long-term consequences [30].

### 1.2. Decision-Making: The Iowa Gambling Task (IGT)

The IGT [10] is one of the most widely used tests to measure decision-making. The IGT is a test that simulates real-life decision-making conditions, where a choice must be made between obtaining a greater immediate reward or another option with which a greater long-term benefit can be obtained. The task requires choosing a card from a set of four decks. In Decks A and B, you receive more money for each card chosen, but you can also lose more money, being disadvantageous in the long run. In contrast, Decks C and D yield half as much money in winnings but less in losses; thus, they are advantageous in the long run. In addition, one of the decks in each group, described by the payoff structure, has a low probability (1 out of 10) of receiving a losing card (Decks B and D), while the other has a high probability (out of 10) of receiving some type of loss (Decks A and C).

According to Bechara et al. [31], the ability to delay rewards lies in the ventromedial prefrontal cortex area; thus, patients who have suffered damage to the ventromedial prefrontal cortex area have severe difficulties successfully performing the IGT task [32]. Hooper et al. [33] analyzed the performance of the IGT in 145 individuals from ages 9 to 17 years and observed that the development of this brain area is closely linked to the decision-making process. In addition, this study found that older adolescents could perform the test more successfully than younger adolescents. This showed that those with less ventromedial prefrontal cortex development made individuals more insensitive to long-term rewards, causing them to commit more risks. Similarly, this long-term reward insensitivity has been observed with the IGT in other types of patients with neuropsychological damage [34], drug addiction problems, pathological gambling, etc. [35,36].

Other studies employing the IGT test in nonclinical populations have observed that adolescents have serious difficulties in sacrificing immediate rewards for more long-term ones [16]. Although young adolescents perform worse on the IGT than healthy adults, several longitudinal studies have observed that IGT performances improve throughout adolescence [17,37]. Prencipe et al. [38], in a study on the development of EF from childhood to adolescence, found that improvements in hot EF, as indicated by one’s performance on the IGT, are observed from late adolescence onward. In contrast to adolescents, young adults tend to make decisions based more on the likelihood of long-term rewards.

### 1.3. EF: Wisconsin Card Sorting Task (WCST)

One of the most commonly used tasks to measure the EF is the WCST [39]. This task is a set-shifting task that involves cognitive flexibility and the ability to maintain and change rules throughout a matching task. The most commonly used measure of executive control in the WCST is the measure of perseverative errors [40]. The perseverative errors total reflects the number of errors a subject makes when maintaining the previous choice category after the rule has changed. There is neuroimaging evidence that the WCST performance is critically associated with the functioning of the dorsolateral prefrontal cortex in healthy subjects [41]. Gläscher et al. [42] found that patients with damage to the dorsolateral prefrontal cortex area had a higher number of perseverative errors in the WCST than healthy subjects.

On the other hand, there is evidence that the WCST performance in healthy individuals varies with age. Thus, Lin et al. [43], in a study with adolescents, showed an improvement with age in the percentages of perseverative errors in the WCST in a sample of 13- to 15-year-olds. The improvement in WCST performance throughout adolescence may be due to the maturation of the dorsolateral prefrontal area [44]. Miranda et al. [45] analyzed a sample of participants from the ages of 18 to 89 and observed that the age of the participants was negatively associated with the number of perseverative errors; thus, age plays a crucial role in the results obtained in the WCST. Rhodes [46], in a study of two meta-analyses, found similar results. In this case, the EF through the WCST was studied in two age groups (20 to 35 years and over 55 years), and it was observed that the older the age was, the less frequently perseverative errors were committed. Rammal et al. [47], after performing the WCST test on 220 individuals aged from 18 to 64 years, found that the older the age, the lower the number of PEs. However, some studies have shown that the improvement in WCST performance reaches its peak during adulthood and that from the age of 60 onward, there is a decreasing trend in WCST performance, with an increase in perseverative errors and a decrease in EF [48].

### 1.4. Decision-Making and EF

In line with the dual model, the interrelationship between the maturation of these two cognitive processes, decision-making and the EF, should be observed throughout adolescence. Some studies have observed a negative correlation between the number of perseverative errors in WCST and IGT performance [49]. Smith et al. [28] further elaborated on this relationship between the WCST and IGT by incorporating the evolutionary dimension. These authors carried out a study with ten age groups, from 8 to 17 years old, and the results revealed both an increase with age in EF, as measured by the WCST, and improved performance on the IGT in older children. These results suggest that the development of EF and risky decision-making have two different but related maturational pathways. Thus, in early adolescence, the immaturity of the EF and impulse control processes would be associated with the selection of the most disadvantageous decks in the IGT, whereas as the frontal lobes mature, the increase in EF allows for better control of the affective impulses and leads to more advantageous decision-making [9,21,22].

However, several studies have not observed this association between the IGT and WCST [50], and in a study of patients with dorsolateral damage, it was found that performance on the IGT and WCST was not related. Adinoff et al. [51], in patients with stimulant use, also found no relationship between the results obtained in the IGT and WCST. Overman [52], in his study comparing ages 5 to 69 years on the differentiation of prefrontal cortex-related tasks, found that there was no difference between the IGT and WCST in subjects aged 5 to 69 years. Finally, Reynolds et al. [53] measured the relationship between the EF measured with the WSCT and decision-making (IGT) in young adults without neurological damage. They found that the WSCT and IGT scores were associated with traits of impulsivity and risk-seeking behavior but did not observe a direct relationship between the WSCT and IGT scores.

This contradiction between the results of studies analyzing the relationship between EF and decision-making may be due to the complexity of the IGT. According to Brand and collaborators, the IGT is a complex test because it is divided into two distinct parts: the first part being ambiguity and the second part being risk [54,55]. The first part is measured by the first trials, where subjects do not have all the information to make their decisions and, therefore, can be considered a test in a situation of ambiguity. In the second part (measured by the latter trials), the subjects have enough information to know the consequences of their decisions, and therefore it becomes a risky task. Almy et al. [14] note that these discrepant findings may be due to correlating EF or intelligence with the IGT overall performance without considering specific task phases. The absence of an effect of EF for decision-making under ambiguity and the presence of an effect for decision-making under risk in the current study speaks to the interpretive utility of dividing the IGT into these two phases [55,56]. Thus, EF may not be related to the total performance in the IGT task [57] but rather only differentially associated with the ambiguous or risky part. The EF scores do not seem to be related to the first block in the IGT but rather to the remaining blocks. When decision-making is risky, low WCST scores were observed to be related to the choice of disadvantageous decks. Similarly, Ouerchefani et al. [32] found that patients with ventromedial or dorsolateral damage performed worse on the EF tasks than healthy adults and scored worse on the IGT task than healthy adults in the risky blocks, corresponding to the last three blocks (Blocks III, IV and V); however, no differences were observed in the ambiguity blocks, corresponding to the first two blocks (Blocks I and II). In turn, Malone et al. [58], in a monozygotic cotwin-control design study, investigated adolescents performing the IGT. The study tested for differences in the adolescent cortical regions for decision-making tasks as a function of alcohol consumption, controlling for genetic characteristics using a twin study. The results showed that alcohol consumption was associated with poorer task performance and a reduced volume of the right orbitofrontal cortex. The study showed that drinkers performed worse on the decision-making task by making more disadvantageous decks, and the difference between drinkers and nondrinkers increased, especially in the last blocks (Blocks IV and V) of the task, where decision-making is risky. More specifically, Almy et al. [11], in a longitudinal study of adolescents, observed that performances on the first two IGT blocks (ambiguity) correlated with age but not with intelligence measures, whereas on the last three IGT blocks (risk), both age and intelligence were positive predictors of advantageous decisions.

According to the literature reviewed, compared to healthy adults, young adolescents perform worse on the IGT task [23], and at least with the healthy adult samples, the findings indicate that EF is associated with risky decisions on the IGT during later blocks. Brand et al. [55], with a sample of healthy adults, did not find a correlation between the first blocks of the IGT, corresponding to Blocks I and II (ambiguous decision-making), and the WCST, whereas they did find that WCST performance was associated with the last blocks of the IGT, which would correspond to Blocks IV and V (risk decision-making). To the best of our knowledge, there is no study among healthy early adolescents and young adults that specifically analyzes the relationship between the WCST and performance in the IGT throughout the task trials. Additionally, there are no conclusive data on whether the results in early adolescence are the same in all blocks of the IGT, and we do not know how these relate to cognitive control processes. Likewise, there are no conclusive results comparing how these relationships change as a function of age during adolescence.

### 1.5. Purpose of the Present Study

This study analyzes the relationship between the IGT decision-making and executive control processes measured through the WSCT at two specific ages: in early adolescence, where more affective impulse processes develop, and in younger adulthood, where cognitive control processes have already matured. Another objective of this study is to analyze how this relationship changes throughout the IGT trials, that is, how, depending on the type of decision-making (ambiguous or risky) occurring, EF processes may play a role in controlling these decisions. Our hypotheses are as follows:Early adolescents will tend to commit more risks in their IGT performance than adults.Given the complexity of the IGT, we hypothesize that the participants will perform worse on the early blocks of the IGT (ambiguous decision-making) while they will perform better on the later blocks (risky decision-making).EF will improve with age; thus, the number of perseverative errors in the WCST will be negatively correlated with the performance in the IGT in adolescents.Controlling for age, we expect that subjects who have fewer perseverative errors on the WCST will tend to make better decisions in the IGT risk blocks.

## 2. Materials and Methods

### 2.1. Sample

The sample of this study comprised 140 people, of whom 70 were early adolescents from first-year secondary school students from a public high school located in a residential area of a medium socioeconomic level. We consider early adolescents as those persons between 12 and 13 years old, which implies a change at a biological, psychological and sociocultural level due to the beginning of their participation in a new educational institution. The mean age of the adolescents was 12.51 (SD = 0.61; 50% women). The group of young adults comprised first-year university students studying for a degree in social work who were enrolled in the subject Introduction to Psychology (mean age: 19.38 years, SD = 1.97; 66% women). We consider young adults as those who are between 18 and 22 years old, which implies that they have new social and legal responsibilities and obligations in Spain.

The sample size was calculated using G*Power 3.1 for a medium effect size, 95% power and *p* < 0.05. The inclusion criterion was that they were freshmen in high school or college, and the exclusion criterion was that they did not have learning difficulties. All the participants agreed to participate, and in the case of minors, signed written consent was requested from their parents. All the procedures were approved by the University Ethics Review Committee for Human Research (number 22/5) and followed the indications of the Council of Europe on Human Rights and Biomedicine in accordance with the 1975 Declaration of Helsinki guidelines

### 2.2. Instruments

#### 2.2.1. Raven’s Progressive Matrices Test

Raven’s Standard Progressive Matrices test [59] was used as a standardized measure of intelligence. Raven’s test comprises five sets of matrices composed of 12 items each, with an increasing level of difficulty. The total Raven’s raw data were centered on the mean for each age group to account for multicollinearity, with age given in years.

#### 2.2.2. Wisconsin Card Sorting Task (WCST)

A computerized version of the WCST, developed using the Psychology Experiment Building Language (PEBL) software [60,61], was used to assess the EF. Following the original WCST procedure [62], four cards are arranged at the top of the screen with different colors, shapes and numbers of items. A new card is presented below on each trial, and the participant must select which of the four sample cards it can pair with one of the following three rules: color, shape or a number of items. The matching rule is maintained until ten correct trials are performed in a row, at which point it switches to the next rule without giving a warning to the participant. The task ends when the set of rules (color, shape and number) has been completed three times or when 128 trials have passed. The sequence of rules was set in pseudorandom order for each subject, with the condition that the same rule was not repeated consecutively. The standard measure of the WCST is the number of perseverative errors, calculated as the sum of responses where the previous classification category is used after it has changed. Since the task could be completed in at least 60 trials, the percentage of perseverative errors over the total number of trials performed was calculated to control for the number of trials. Additionally, the percentage of perseverative errors was centered on the mean for each age group to account for multicollinearity, with age given in years.

#### 2.2.3. Iowa Gambling Task

In this study, a computerized version of the Iowa Gambling Task [10] was employed using Inquisit 3.5 software. In the IGT, four cards are presented in the center of the screen, labeled A, B, C and D; each time the participant chooses a card, he can obtain a benefit or a cost, and the payoff structure is different for each pair of cards. Choosing a card from Decks A and B always results in an immediate prize of 100 Euros, while Decks C and D yield a prize of 50 Euros. However, in Decks A and B, every 10 selections of a card resulted in a total cost of 250 Euros; in contrast, Decks C and D, in the same number of selections, resulted in a total profit of 250 Euros. Furthermore, the decks differ in the frequency of punishments: Decks A and C have a high probability of receiving a losing card (5 out of 10), and Decks B and D have a low probability of taking a losing card (1 out of 10). The participants were previously unaware of the payoff structure of the decks. The only thing they were informed about was that the objective of the task was, starting from 2000 Euros, to obtain as much money as possible in 100 trials. During the task, the computer provided visual information via text and by means of a bar graph of the money won and lost in each trial and of the progress in the total count. To evaluate performance in the IGT, we usually use the net score calculation, resulting from subtracting the number of choices in the advantageous decks (C + D) from the number of responses in the disadvantageous decks (A + B). To analyze the effect of the frequency of losses, the difference between the number of choices in the decks with a high frequency of losses (B + D) and those with a low frequency of losses (A + C) was also calculated. The overall net scores across blocks of 20 trials were computed.

### 2.3. Procedure

Data collection was divided into two days within the range of one week. On the first day, Raven’s test was administered collectively in a class of 20 students by two assistant researchers. Each participant was given a handout with the matrices and a sheet to record the answers. They were given as much time as they needed to complete the test. The computerized tasks were carried out in a room set up for data collection in the educational center. Four computers were distributed throughout the room so that each participant could not see the others and with enough space so they could not communicate with each other. All participants received the same instructions before the tasks. The instructions were displayed in text on the computer, and an assistant researcher remained in the room until the tasks were finished. The order of the tasks was randomly assigned by the computer; there was no significant effect of the task order on any measure analyzed.

All the procedures of this study were approved by the University Ethics Review Committee for Human Research (number 22/5) and followed the indications of the International Conference of Good Clinical Practices in accordance with the 1975 Declaration of Helsinki guidelines.

## 3. Results

### 3.1. Performance in Iowa Gambling Task

A repeated measures analysis of variance was performed with Deck (4) as the within-subject variable and Age Group (2) X Gender (2) as the between-subject variables. The results with the Greenhouse–Geisser correction, for violation of the sphericity assumption, indicated that there were significant differences between the decks: F(2.016, 274.968) = 79.521; *p* < 0.001; Eta = 0.369. The Bonferroni-adjusted post hoc tests revealed that decks with a low frequency of misses, Deck B (M = 31.57; SD = 12.26) and Deck D (M = 32.09; SD = 9.34), were significantly more selected than the decks with a high frequency of misses, Deck A (M = 17.88; SD = 6.61) and Deck C (M = 18.41; SD = 7.51), ps < 0.001. However, there were no significant differences within Decks B and D or within Decks A and C. There was an interaction effect of age group and deck selection: F(2.022, 274.968) = 3.170; *p* < 0.05; Eta = 0.023. The Bonferroni-adjusted post hoc tests revealed that the young adults selected significantly more cards from the advantageous Deck D (M = 34.35; SD = 9.20) than the early adolescents (M = 29.82; SD = 8.98), *p* < 0.01. There were no significant differences by gender or interaction effect across the decks, age groups and gender.

In order to analyze the differences observed between the consequences in the long term and the frequency of penalties, a repeated measures analysis of variance (ANOVA) was performed with Consequences (2) X Frequency of Punishment (2) as the within-subjects factors and Age Group (2) X Gender (2) as the between-subjects factors, following the Greenhouse–Geisser correction. As described before, the decks with a lower rate of punishment were selected more (M = 31.83, SD = 6.45) than those with a higher frequency of penalty cards (M = 18.15, SD = 6.47); F(1, 136) = 156.622; *p* < 0.001; Eta = 0.535. There were no interaction effects across the frequency of punishment, age or gender. However, there was a significant interaction effect between the deck’s long-term consequences and age groups: F(1, 136) = 6.184; *p* < 0.05; Eta = 0.043. The early adolescents selected significantly more cards from the disadvantageous decks (M = 25.76, SD = 5.10) than the young adults (M = 23.69, SD = 4.25); *p* < 0.05, whereas the young adults selected significantly more cards from the advantageous decks (M = 26.27, SD = 4.26) than the early adolescents (M = 24.24, SD = 5.10); *p* < 0.05. Additionally, the young adults drew more cards from the advantageous Decks C and D than from the disadvantageous Decks A and B, *p* < 0.05, whereas there was no such difference in the total amount of cards picked by the profit of decks among the early adolescents.

Following the previous studies, the IGT’s performance measure was divided into five blocks of 20 card selections to examine the changes in performance over time. Within each block, an overall net score was computed as the number of advantageous (C + D) minus disadvantageous (A + B) choices. A repeated measures analysis of variance (ANOVA) was performed using a within-subjects factor Task Block (5) X Age Group (2) X Gender (2) as the between-subjects factors, and a Greenhouse–Geisser degrees of freedom correction was reported. The results indicated an increase in advantage choice behavior across the task blocks; the main effect of the task blocks was as follows: F(4, 496.476) = 17.975, *p* < 0.001, Eta = 0.117. The performance also increased with age; the main effect of the age group was as follows: F(1, 136) = 6.184, *p* < 0.05, Eta = 0.043; see Figure 1. However, there was no significant interaction between the task blocks and age groups. There was no significant main effect of gender: F(1, 136) = 0.04, ns; there were also no significant interaction effects among genders, age groups and task blocks, with all being a *p* > 0.05. A Bonferroni-adjusted multiple pairwise comparison post hoc test indicated that young adults selected more advantageously (M = 2.33, SD = 5.96) than early adolescents (M = −0.085, SD = 5.79) in the last block (*p* < 0.05) but there were no significant differences between the age groups in the previous blocks.

In order to assess our hypothesis about the differences between ambiguous decision-making (the first and second blocks) and risky decision-making (the third to fifth blocks) within the IGT task, a repeated measures analysis of variance (ANOVA) was performed with the decision-making block (2) as within-subjects factors and Age Group (2) X Gender (2) as the between-subjects factors, following the Greenhouse–Geisser correction. There was a significant difference in the decision-making blocks; among all participants, the mean net scores were lower on the ambiguity blocks (M = −1.17, SD = 4.32) than on the risky blocks (M = 1.13, SD = 4.39); F(1, 136) = 39.430; *p* < 0.001; Eta = 0.225. There was a significant main effect of group age; the young adults received higher net scores on the ambiguity and risky blocks than the early adolescents; F(1, 136) = 6.102; *p* < 0.05; Eta = 0.043. A Bonferroni-adjusted multiple pairwise comparison post hoc test indicated that young adults selected more advantageously than early adolescents as much as in the ambiguity blocks, M = −0.43, SD = 3.93 and M = −1.91, SD = 4.59, respectively, *p* < 0.05, as in the risky blocks, M = 2.00, SD = 4.38 and M = 0.26, SD = 4.26, respectively, *p* < 0.05. There was no main effect of gender or interaction effects across the decision-making blocks, age or gender.

### 3.2. Performance in Cognitive Tasks

The Gender (2) X Age Group (2) analyses of variance (ANOVA) revealed that the overall score in Raven’s task increased with age, F(1, 136) = 30.754, *p* < 0.001, Eta = 0.184; see Table 1. However, the percentage of perseverative errors on the WCST decreased significantly with age, F(1, 136) = 13.591, *p* < 0.001, Eta = 0.091; see Table 1. There was no significant main effect of gender or interaction between age and gender on the Raven and Wisconsin tasks. As expected, there were nonsignificant differences by age group when the group-mean-centering scores of Raven’s task and the WCST were used as the dependent variables.

### 3.3. IGT Measures Associated with Performance in Cognitive Tasks

Correlational analyses of age in years, gender (female = 1) and cognitive measures with the IGT performance scores are displayed in Table 2. Age in years was positively correlated with a Raven total score (r = 0.443, *p* < 0.01) and negatively correlated with the percentage of perseverative error of the WCST (r = −0.344, *p* < 0.01). The nonsignificant correlations between Raven’s test total scores and the perseverative error measure of the WCST are as follows: r = −0.076, *p* = 0.37. The Raven’s scores were not correlated with the total number of cards picked on any deck from the IGT; all *p*s were n.s. However, there was a negative correlation between the percentage of perseverative errors on the WCST and the number of cards picked from Deck C, r = −0.21, *p* < 0.05, and a positive correlation with the number of cards picked from Deck B, r = 0.20, *p* < 0.05. Thus, those who committed more perseverative errors on the WCST were less likely to pick cards from the advantageous deck with a higher frequency of punishments and were more likely to pick cards from the disadvantageous Deck B with a lower frequency of punishments. 

Correlational analyses revealed nonsignificant correlations between the IGT total net score and Raven’s task measures (see Table 2). However, the total IGT net score was significantly negatively correlated with the percentage of perseverative errors on the WCST, r = −0.222, *p* < 0.01. Even so, when the IGT net score was introduced by blocks, the percentage of perseverative errors on the WCST was negatively correlated with the IGT net score on the third block, r = −0.180, *p* < 0.05, and the fifth block, r = −0.298, *p* < 0.01. Furthermore, the percentage of perseverative errors was not significantly correlated with the IGT net score for the ambiguous decision-making blocks (first and second), but it was negatively correlated with the net score for the IGT’s risky decision-making blocks (third to fifth), r = −0.258, *p* < 0.01. Raven’s total score was positively correlated with the IGT net score on the second block, r = 0.17, *p* < 0.05.

### 3.4. Regressions

Regression analyses were used to assess whether the executive measures could account for the overall performance of the IGT when controlling for the age in years, gender and fluid intelligence (see Table 3). Given the high multicollinearity between age and the scores of the cognitive tasks, the regression models were computed with the group-mean centering WCST´s percentage of perseverative errors and Raven’s total scores. When the total IGT net score was used as the dependent variable, age in years was the only significant predictor: R2 = 0.102, F(4, 135) = 3.852, *p* < 0.01. It was found that when controlling for the rest of the variables, the IGT total net score increased significantly with age (β = 0.246, *p* < 0.01).

Following our hypothesis about the differences between ambiguity and risky blocks within the IGT, regression analyses were used with the mean net scores of the two groups of decision-making blocks used as dependent variables; see Table 3. The regression model was nonsignificant for the mean net scores of the ambiguity blocks (first and second), R2 = 0.056, F(4, 135) = 1.992, *p* > 0.05; even so, age in years was the only significant predictor that accounted for the mean IGT net score of the first two blocks, the ambiguity task. However, in the risky IGT blocks (third to fifth), age in years and the percentage of perseverative errors were significant predictors of the IGT mean net score, R2 = 0.106, F(4, 135) = 4.012, *p* < 0.01. The data show that the mean net score increased significantly with age by as much in the ambiguity blocks (β = 0.202, *p* < 0.05) as it did in the risky blocks (β = 0.226, *p* < 0.01). However, in the risky blocks, the percentage of perseverative errors negatively predicted the mean net score (β = −0.189, *p* < 0.05), showing that those who committed more perseverative errors in the WCST were more likely to take cards from the disadvantageous decks in the last blocks of the IGT.

## 4. Discussion

Adolescence is usually observed as a period when young people tend to make more risky decisions [2]. With age, adolescents become more aware of the long-term consequences of their actions and thus become increasingly risk-averse [29,63]. This study analyzed whether differences in risky decision-making in early and late adolescence are related to changes in EF. The development of EF increases substantially in early adolescence and is relatively mature by early adulthood. Thus, it is expected that there is a relationship between this development of EF and the reduced tendency to make risky decisions throughout adolescence into adulthood.

The data obtained in this study confirm our first hypothesis: early adolescents took more risks than young adults in the IGT. Young adults selected more cards from the long-term advantageous decks than from the disadvantageous decks, whereas no such differences in payoffs were observed among adolescents. In addition, in total, young adults selected more cards from the advantageous Deck D, with a low loss frequency, than adolescents. These results are similar to previous studies that observed an increase in advantageous long-term decisions with age [38].

However, in all cases, for both adolescents and young adults, the participants in this study selected more cards from the low loss frequency Decks B and D than from the high loss frequency Decks A and C, regardless of the total amount to be won or lost. Previous studies found that healthy participants tend to select more cards from decks with a lower frequency of losses in the IGT [32]. As has been repeatedly mentioned, the punishment frequency of decks of cards can interfere with the learning of long-term gains and losses. Both healthy and clinical subjects of all ages tended to choose more cards from the decks with a low punishment frequency, regardless of whether they were advantageous or disadvantageous [64]. Previous studies using the Two-Alternative-Forced-Choice (TAFC) tasks have shown a similar bias for the most immediate option on decisions regarding risk perception, especially when associated with cognitive deficits [65].

According to the approach of our second hypothesis, a block analysis of the IGT showed that in all age groups, learning of the consequences of each deck was observed throughout the performance of the task. All participants demonstrated a pattern of learning the long-term consequences of their choices, with the frequency of selections of advantageous decks per block progressively increasing. Moreover, when the blocks were divided between ambiguity (Blocks I and II) and risk (Blocks III to V), there was a significant difference in the decision-making blocks; among all the participants, the selection of advantageous decks of cards was lower for the ambiguity blocks than the risky blocks. However, there was a significant main effect of group age; the young adults received higher advantageous net scores on the ambiguity and risky blocks than the early adolescents. These data demonstrate that there is indeed learning on the part of both groups during the test. These data are consistent with the study by Almy et al. [14], in which performance on the IGT test improves with age, but at the beginning of the task, adults perform better on the ambiguity blocks than adolescents. This improvement in performance is linear, which makes the results of adults better than those of adolescents, confirming the second hypothesis.

In conclusion, at the beginning of the task, with the ambiguity blocks, adolescents are driven by the immediate consequences of each deck and quickly focus on the magnitude and frequency of short-term rewards. In turn, young adults in the ambiguity block explore the different decks to obtain a more accurate prediction of the long-term consequences of their decisions, whereas in the final blocks, by previous trial and error experience, they are more aware of the payoff structure of each deck. These data are congruent with the dual development theory that states that adolescents are myopic to the long-term consequences of their decisions because of the activation of the brain centers related to pleasure in the presence of an immediate reward and, at the same time, the immaturity of the frontal lobes in charge of behavioral inhibition.

Correlation analyses show, in the total sample, a negative relationship between the number of perseverative errors in the WCST task and IGT performance, confirming the third hypothesis. The deck correlation analysis shows that those participants who made more perseverative errors in the WCST were less likely to select cards from Deck C, which is advantageous in the long term but has a low frequency of reinforcement in the short term. This shows that, in general, the attraction of short-term incentive frequency is superior to the amount of gains obtained in the long term. In Deck C, the amount gained in the long run, after every 10 cards chosen from the same deck, is the same as in Deck D, but the frequency of inducements is higher in Deck D (9 out of 10) than in Deck C (5 out of 10). Thus, although the amount gained in the long run is the same in both cases, yielding a positive gain–loss differential, those participants who demonstrated a worse level of EF in the WCST, i.e., a high frequency of perseverative errors, were more likely to choose based on the frequency of incentives and, consequently, were less likely to take cards from Deck C. In conclusion, those participants who have a lower EF ability exhibit accentuated reward-seeking behavior, regardless of the amount of gains obtained in the long run.

However, the relationship between EF and performance on the IGT varied by block, according to the fourth hypothesis. The correlation analysis of the IGT net score by block and the perseverative errors on the WCST showed that the number of perseverative errors correlated negatively with the IGT net score on the last block. Furthermore, the perseverative errors were not significantly correlated with the IGT net score for the ambiguous decision-making blocks (first and second), but they were negatively correlated with the net score for the IGT’s risky decision-making blocks (third to fifth). In sum, those participants with executive deficits, as shown by the higher number of perseverative errors, performed worse on the IGT in the risky blocks, with learned knowledge of the payoff structure of the decks.

Furthermore, according to our hypothesis about the differences between the ambiguity and risky blocks within the IGT, the regression analyses showed that age in years was the only significant predictor that accounted for the mean net scores of the ambiguity blocks (first and second). However, in the risky IGT blocks (third to fifth), age in years and the percentage of perseverative errors were significant predictors. The mean net score increased significantly with age and was as much for the ambiguity blocks as for the risky blocks. However, in the risky blocks, the percentage of perseverative errors predicted the mean net score, showing that controlling for age among other variables, those who committed more perseverative errors in the WCST were more likely to take cards from the disadvantageous decks on the risky blocks of the IGT. This block analysis shows the importance of learning throughout the IGT test, resulting in a greater difference between adolescents and young adults than if only the final score were taken, as has been the case in previous studies [50]

Almy et al. [14] found similar results in a longitudinal study of healthy adolescents from 9 to 23 years of age. The data showed a positive linear relationship between age and total IGT performance. However, age and cognitive control predicted total performance on the IGT in the risk blocks but not in the ambiguity blocks. Following Almy et al. [14], according to the dual system theory, these results may be due to the differential developmental effect of the centers linked to dopamine and cognitive control in adolescence. This dual system theory proposes that in adolescence, a linear development of cognitive control (EF) and a curvilinear development of sensation-seeking and attraction to reinforcement coexist, whereby for a period of time, adolescents do not have sufficient cognitive control capacity and are overcome by the affective intensity of the incentive. This could be explained by the immaturity of the cognitive control brain centers in the face of overstimulation of the amygdala and ventral nuclei [26,66].

While the dual system theory may account for the differences between adolescents and young adults in the total IGT, our study shows that EF is significantly predicted, controlling for age and IGT performance in the last risk block; the score in the last block was higher for those who made a lower number of perseverative errors in the WCST. In the risk blocks (IV and V), those participants with low EF scores were more likely to select decks with a high incentive frequency, regardless of the number of wins or losses. Thus, they prefer decks with a high probability of obtaining a reward, whether or not it pays off in the long run. That is, participants with lower EFs use a biased decision, focusing only on the frequencies of rewards and penalties to make their decisions, and do not consider the amounts obtained in the short and long term, thus selecting cards from Decks B and D as a priority. However, participants with a higher EF capacity jointly attend to these two attributes, quantity and frequency, elaborating more complex gits; thus, they prefer the deck with a high frequency of rewards but which, in turn, yields a better performance of cumulative quantity between the gains and losses, such as in Deck D. Thus, the EF predicts a better IGT performance in the last blocks, when decision-making is risky; in these blocks, it is necessary to inhibit the selection response of the decks with a higher immediate incentive, to have in working memory, the long-term objective and to be able to focus attention in each round of selection on the long-term consequences obtained and the learning experience of previous trials.

Thus, the increase in EF throughout adolescence could explain the differences between young adults and adolescents in risky decision-making. Adolescent decision-making is biased by motivational factors and incentive-seeking rather than by cognitive calculations of outcome probabilities. Luciana and Collins [67] proposed a brain-based model for understanding adolescent-biased behavior. According to this model, the dopaminergic substrates of incentive motivation increase in adolescence and decrease later in life. This increase interacts with the executive control systems affecting self-regulation. Because of the high task demands on executive control processes due to the increase in dopamine-triggered incentive motivation, adolescents are overloaded in their self-control abilities. Following this model, adolescents would choose on the basis of an expected value, positive or negative, and thus are biased by the frequency of immediate reinforcements, whereas adults select by better integrating their emotional states and giving greater weight to the long-term consequences and cost–benefit balance.

A limitation of this study is that it is based on a cross-sectional design with only two age ranges. Following the dual system theory, this study provides experimental data on the relationship between the adolescent development of executive function and risky decision-making. However, it remains to be clarified whether executive function deficits can predict the prevalence of risky behaviors in the future. Future research would benefit from a longitudinal analysis of the relationship between risky decision-making and the development of EF throughout adolescence. The follow-up of this relationship from early adolescence to early adulthood would allow us to contrast whether, as predicted by the dual system theory, the progressive development of the frontal lobes throughout these ages is reflected in greater executive control and improvement in the capacity to make risky decisions, inhibiting preferences for immediate reinforcement in favor of the long-term benefit. Furthermore, according to Fisk and Sharp [68], executive functioning can be subdivided into four operations: updating, shifting, inhibition and access. Previous studies have shown that age is associated with the type of operations and that the four subcomponents of EFs are associated differently with cognitive abilities, such as the Theory of Mind [69]. In future research, it should be explored which EF operations are associated with risk behavior.

## 5. Conclusions

In our daily lives, we are faced with decisions where we know we run a certain risk of negative or positive consequences in the long term. However, it is not always the case that a decision must be made where the decision-maker knows the possible payoffs; these are ambiguous decisions. Adolescents and adults must face situations where they must make decisions in situations of ambiguity, but other decisions are risky. Adolescents and adults solve ambiguous decisions by trial and error; however, in situations of risk, adolescents are more likely to make decisions without attending to the long-term consequences. The IGT, as used in this study, simulates these ambiguous and risky decision-making situations. According to the dual development theory, in the ambiguity blocks of the task, adolescents tend to select the short-term rewards preferentially, whereas young adults will explore the different options until finding out the long-term consequences of each choice. However, in the risky blocks, where the cost–benefit structure is known by the participants, those with lower executive function skills, regardless of age, are more likely to make selections biased by a high incentive frequency. The current study shows that the EF partly explains this difference in decision-making between adolescents and adults. In turn, in all age ranges, greater executive control abilities are associated with more advantageous decisions in risky circumstances. These data can be used to develop intervention or prevention programs for risky behavior among adolescents. In order to reduce risky decision-making, the development of EF could be promoted with training in inhibition, working memory and focused attention. By taking into account the complexity of the adolescent stage, this research shows that a deficit in EF may be an indicator of difficulties in the management of emotions at an early age and, thus, in risky decision-making. Therefore, the assessment of EF could be a predictor of risk-taking tendencies in decision-making. This finding is of interest for future research that can analyze whether adolescents and young adults who present behavioral problems and risk behaviors, such as substance use or antisocial behaviors, are associated with EF deficits.

## Figures and Tables

**Figure 1 behavsci-13-00142-f001:**
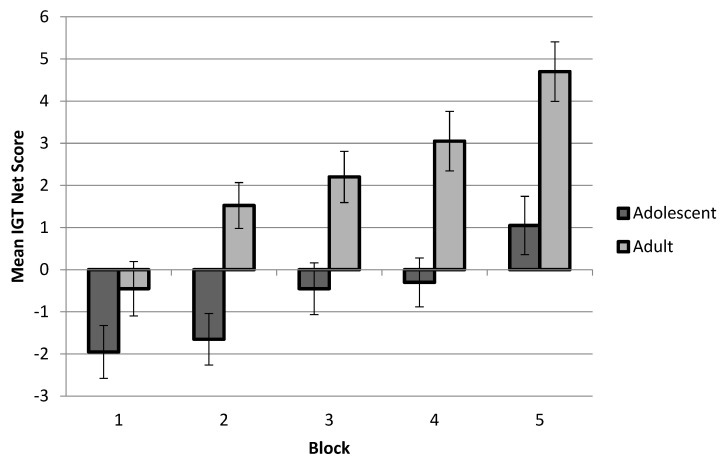
Mean IGT net scores by blocks. Note: mean (±SE) IGT net scores of adolescents and young adults by blocks.

**Table 1 behavsci-13-00142-t001:** Performance means (and standard deviations) for cognitive tasks by age group and gender.

	Adolescence		Young Adult	
	Male	Female	Male	Female
Raven score	41.05 (8.12)	40.58 (7.16)	48.50 (6.03)	45.89 (5.51)
WCST perseverative errors %	18.81 (7.02)	20.21 (8.04)	15.88 (7.89)	14.33 (5.14)
IGT Deck A	17.94 (7.41)	18.69 (6.74)	16 (6.19)	18.71 (5.92)
IGT Deck B	34.16 (16.26)	32.18 (11.49)	32.78 (12.04)	27.50 (6.84)
IGT Deck C	18.43 (8.76)	18.87 (7.42)	17.59 (7.52)	18.68 (6.41)
IGT Deck D	29.45 (9.82)	30.24 (8.07)	33.62 (7.63)	34.97 (10.04)
IGT Total net score	−0.84 (0.61)	−0.35 (0.65)	0.48 (0.66)	1.48 (0.61)
IGT net score Blocks 1–2	−2.08 (5.37)	−1.72 (3.59)	−0.71 (4.97)	−0.18 (2.79)
IGT net score Blocks 3–5	−0.01 (4.71)	0.56 (3.72)	1.29 (4.69)	2.60 (4.07)

Note: IGT = Iowa Gambling Task, WCST = Wisconsin Card Sorting Task.

**Table 2 behavsci-13-00142-t002:** Correlation analyses between cognitive tasks and IGT measures.

	IGT Scores											
Measure	1	2	3	4	5	6	7	8	9	10	11	12
Age in years	−0.062	−0.174 *	0.008	0.263 **	0.133	0.229 **	0.178 *	0.175 *	0.227 **	0.212 *	0.245 **	0.264 **
Sex (1 = Female)	0.126	−0.157	0.049	0.074	−0.026	0.140	0.070	0.152	0.070	0.064	0.122	0.113
Raven score	−0.074	−0.051	−0.010	0.131	0.079	0.170 *	−0.021	0.079	0.116	0.145	0.077	0.118
WCST perseverative errors %	−0.054	0.203 *	−0.210 *	−0.058	−0.090	−0.089	−0.180 *	−0.116	−0.298 **	−0.106	−0.253 **	−0.222 **

Note: IGT = Iowa Gambling Task, WCST = Wisconsin Card Sorting Task; 1. Deck A total score; 2. Deck B total score; 3. Deck C total score; 4. Deck D total score; 5. Block 1 net score; 6. Block 2 net score; 7. Block 3 net score; 8. Block 4 net score; 9. Block 5 net score; 10. Blocks 1–2 net score; 11. Blocks 3–5 net score; 12. Total net score. * *p* < 0.05. ** *p* < 0.01.

**Table 3 behavsci-13-00142-t003:** Summary of regression analysis for variables predicting net score. Net Score Blocks 1–2, Net Score Blocks 3–5 and Net Score Total on the Iowa Gambling Task (N = 140).

	IGT Net Score Blocks 1–2			IGT Net Score Blocks 3–5			IGT Total Net Score		
	B	SE B	β	B	SE B	β	B	SE B	β
Age	0.240	0.100	0.202 **	0.274	0.099	0.226 **	0.260	0.087	0.246 **
Sex1	0.503	0.727	0.058	0.927	0.719	0.106	0.758	0.629	0.099
Raven total score	0.051	0.054	0.080	0.003	0.053	0.005	0.022	0.047	0.039
WCST percentage of perseverative errors %	−0.030	0.052	−0.049	−0.118	0.051	−0.189 *	−0.083	0.045	−0.152

Note: IGT = Iowa Gambling Task, WCST = Wisconsin Card Sorting Task. 1 Female = 1. * *p* < 0.05. ** *p* < 0.01.

## Data Availability

The datasets used in the current study are available from the authors upon reasonable request and with the permission of all authors.

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
