# Peer review of "Risk Decision Making and Executive Function among Adolescents and Young Adults"

_behavsci, 2023, doi:10.3390/bs13020142_

Round 1

Reviewer 1 Report

Since the age of the participants has been considered as a variable, thus it is advisable to mention age in the abstract.

Mention the future scope of the work.

What exclusion/inclusion criteria were followed to recruit the participants?

Please check the font "In our daily lives, we are faced with decisions where we know we run a certain risk 580 of negative or positive consequences in the long term."

The conclusion needs to be rewritten where it should reveal the outcome of the paper, not the generalized concepts.

Few of the sentences seems directly copied from the source, such as "In our daily lives, we are faced with decisions where we know we run a certain risk 580 of negative or positive consequences in the long term." Please cross check their content.

Cite a few of the references giving a glimpse of the TAFC tasks utilized in recent papers:

[]Wadhera, T., & Kakkar, D. (2021). Modeling risk perception using independent and social learning: application to individuals with autism spectrum disorder. The Journal of Mathematical Sociology45(4), 223-245.

Author Response

Thank you very much for reviewing our paper, behavsci-2181319, entitled “Risk Decision Making and Executive Function Among Adolescents and Young Adults”, and for giving us the opportunity to submit a revised revision for further consideration in the Behavioral Science. We would like to thank the reviewers for their helpful comments on the manuscript. Below we reproduce all of the reviewers’ comments in bold and place our responses below in regular text.

Since the age of the participants has been considered as a variable, thus it is advisable to mention age in the abstract.

Thanks for the comment, the abstract has been modified to include the age of the participants.

“This study analyzes the relationship between decision-making and EF processes in a group of early adolescents (mean age = 12.51 years, SD = 0.61), where more affective impulse processes are developed, and in young adults (mean age = 19.38 years, SD = 1.97), where cognitive control processes have already matured.” (Lines 11-15)

Mention the future scope of the work.

Thank you for indicating this point, in order to include the contribution in future studies, the following is added in the conclusions section:

“This finding is of interest for future research that can analyze whether adolescents and young adults who present behavioral problems and risk behaviors, such as substance use or antisocial behaviors, are associated with EF deficits.” (Lines 629-632)

What exclusion/inclusion criteria were followed to recruit the participants?

Thank you for pointing this out, the inclusion and exclusion criteria are now described in the Participants section of the manuscript:

“The inclusion criterion was that they were freshmen in high school or in college, and the exclusion criterion was that they did not have learning difficulties.” (Lines 254-255)

Please check the font "In our daily lives, we are faced with decisions where we know we run a certain risk of negative or positive consequences in the long term."

Thank you for your comment. However, this sentence was written by the authors in an original way, even so we have made a check if it can overlap with other publications using SafeAssign and Turnitin (plagiarism detection software) and the result obtained is that there is no coincidence with previously published works.

The conclusion needs to be rewritten where it should reveal the outcome of the paper, not the generalized concepts.

Thank you for pointing out the need to disclose specific results in the conclusion of the manuscript, following the recommendation the following has been included in the paragraph:

The IGT, as used in this study, simulates these ambiguous and risky decision-making situations. According to the dual development theory, in the ambiguity blocks of the task, adolescents tend to select preferentially the short-term rewards, whereas young adults will explore the different options until finding out the long-term consequences of each choice. However, in the risky blocks, where the cost-benefit structure is known by the participants, those with lower executive function skills, regardless of age, are more likely to select biased by high incentive frequency.” (Lines 614-620)

Few of the sentences seems directly copied from the source, such as "In our daily lives, we are faced with decisions where we know we run a certain risk 580 of negative or positive consequences in the long term." Please cross check their content.

Thank you for your comment. However, the manuscript has been written by the authors taking care to include the references of all citations and avoiding any coincidence with previous published texts that could give rise to suspicions of plagiarism. However, we have checked whether the text of the manuscript may overlap with other publications using SafeAssign and Turnitin (plagiarism detection software) and the result obtained is that there is no overlap with previously published work.

Cite a few of the references giving a glimpse of the TAFC tasks utilized in recent papers:

[]Wadhera, T., & Kakkar, D. (2021). Modeling risk perception using independent and social learning: application to individuals with autism spectrum disorder. The Journal of Mathematical Sociology, 45(4), 223-245.

Thank you very much for this insightful comment, we did not know the TAFC tasks and after a review of the literature we have found that there are similarities with the IGT task and the results of previous studies with TAFC task are interesting to interpret the risk decisions. This is now commented in the present manuscript in the discussion section as follows:

“Previous studies using Two-Alternative-Forced-Choice (TAFC) tasks have shown a similar bias for the most immediate option on decisions regarding risk perception, especially associated with cognitive deficits [65].” (Lines 481-483)

We have entered the new reference [65] in the reference section of the manuscript:

Wadhera, T., & Kakkar, D. (2021). Modeling risk perception using independent and social learning: application to individuals with autism spectrum disorder. J Math Sociol, 45, 4, 223-245. https://doi.org/10.1080/0022250X.2020.1774877

Reviewer 2 Report

This is a very interesting and well-done paper and I have some small comments for the authors.

First, I would like to see a clear definition of developmental stages for early adolescents and young adults, how and why selected and how relevant for this study and overall to advance novel insights. The methods section mentions student samples but these are not the same as adolescent and young adults. What about emerging adults?

Second, I miss a section on the country and the sample, what we know from past work there and how relevant is the cultural context. I would assume these are Spanish samples but what we know about adolescents and young adult there?

Third, please add a convincing text on the incremental contributions of this work globally and in Spain, developmentally and for the specific groups investigated.

How were the measures adapted to Spanish and any evidence of reliability?

Please avoid too short and single sentence paragraphs, or too long paragraphs as hard to digest.

Author Response

Thank you very much for reviewing our paper, behavsci-2181319, entitled “Risk Decision Making and Executive Function Among Adolescents and Young Adults”, and for giving us the opportunity to submit a revised revision for further consideration in the Behavioral Science. We would like to thank the reviewers for their helpful comments on the manuscript. Below we reproduce all of the reviewers’ comments in bold and place our responses below in regular text.

This is a very interesting and well-done paper and I have some small comments for the authors.

We appreciate your feedback and will try to respond to your reviews to the best of our ability.

First, I would like to see a clear definition of developmental stages for early adolescents and young adults, how and why selected and how relevant for this study and overall to advance novel insights. The methods section mentions student samples, but these are not the same as adolescent and young adults. What about emerging adults?

Thank you very much for your very helpful comment, we have introduced a clearer and more precise definition of early adolescent and younger adulthood in the sample section:

We have chosen early adolescents (lines 81-90), because it is a stage in which FE and control processes are immature. However, reward processing based on sub-cortical brain structures (amygdala) is quite developed.

Young adults have been selected because at this stage FE and control processes linked to the maturation of the neurological cognitive-control system,  are in the final stage of maturation or close to it, so they are able to regulate and inhibit emotional impulses, which are also mature.

The manuscript does not use the term of emerging adults, because we focused on the developmental stage of the life span, in which EF that regulates emotional processes are maturing; whereas emerging adults refers more to the sociocultural background and changes. in this period of transition from one stage of the life cycle to another, the EF that regulates emotional processes are not yet mature. 

We explain and describe this as follows now in the Sample section:

“The sample of this study consisted of 140 people, of whom 70 were early adolescents from first-year secondary school students from a public high school located in a residential area of medium socioeconomic level. We have considered early adolescents as those persons between 12 and 13 years old, which implies a change at a biological, psychological and socio-cultural level, due to the beginning of their participation in a new educational institution. The mean age of the adolescents was 12.51 (S.D. = 0.61; 50% women). The group of young adults consisted of first-year university students in the degree of social work who were enrolled in the subject Introduction to Psychology (mean age: 19.38 years, S.D. = 1.97; 66% women). We have considered young adults those who are between 18 and 22 years old, which implies that they have new social and legal responsibilities and obligations in Spain.” (Lines 241-252)

Second, I miss a section on the country and the sample, what we know from past work there and how relevant is the cultural context. I would assume these are Spanish samples but what we know about adolescents and young adult there?

We find very interesting the option of introducing a section with the characteristics of the sample in the country of the study, Spain. We appreciate the gap you pointed out and try to give some cultural context in the introduction as follows:

“Adolescence is a stage of increased vulnerability, in which there is an accelerated transition of physical, biological, intellectual, and social changes, which raises the probability of risky behaviors such as the use of illegal substances. the Survey on Drug Use in Secondary Education in Spain [5] has reported alarming rates of alcohol use (53.6%), tobacco use (23.9%) and marijuana use (23.9%) use in Spanish adolescents, aged 14 to 18 years, in the last 30 days. Espada et al. [6] observed in a sample of Spanish adolescents aged 15-18 years an increase in high-risk sexual practices between 2006 and 2012. Similar results have recently been observed in relation to risky behaviors on the road in young people in Spain [7].” (Lines 35-43)

 We have entered the new references [5-7] in the reference section of the manuscript.

Observatorio Español de las Drogas y las Adicciones. Alcohol, tabaco y drogas ilegales en España. Ministerio de Ssanidad: Madrid, Spain, 2022.

Espada, J.P., Escribano, S., Orgilés, M., Morales A. & Guillén-Riquelme, A.  (2015) Sexual risk behaviors increasing among adolescents over time: comparison of two cohorts in Spain. AIDS Care, 27, 6, 783-788, doi: 10.1080/09540121.2014.996516

Alonso, F., Esteban, C., Useche, S., Colomer, N. Effect of Road Safety Education on Road Risky Behaviors of Spanish Children and Adolescents: Findings from a National Study.  J. Environ. Res. Public Health, 2018, 15, 2828. https://doi.org/10.3390/ijerph15122828

Third, please add a convincing text on the incremental contributions of this work globally and in Spain, developmentally and for the specific groups investigated.

Thank you for indicating this point, our main findings support the dual system theory but are limited to a cross-sectional study. Longitudinal studies should be done in future researches to follow up the relations between EF and risk behavior.  In order to include the contribution of this work and pointing out to future studies, the following is added in the discussion section:

“Following the dual system theory, this study provides experimental data on the relationship between adolescent development of executive function and risky decision making. However, it stays to be clarified whether executive function deficits can predict the prevalence of risky behaviors in the future. ” (Lines 591-594)

 How were the measures adapted to Spanish and any evidence of reliability?

Thank you for your comment, the tests and tasks used in the research meet the standards of validity and reliability for the Spanish population.

Raven's Progressive Matrices Test is a standardized intelligence test adapted to the Spanish population (Sánchez-Sánchez, F., Santamaría, P. and Abad, F. J. (2015). Matrices. Test de Inteligencia General. Madrid: TEA Ediciones)Wisconsin Card Sorting Task (WCST) is a neuropsychological assessment test of several components of the executive functions, such as abstract reasoning, category formation, problem solving and perseveration published in tea editions and adapted for the Spanish population by M. V. de la Cruz (R&D Dept. of Hogrefe TEA Ediciones).

IGT has been used in a previous study in a sample of university students in 2015 where they found high levels of reliability: Alarcón, D., Amián, J. G., & Sánchez-Medina, J. A. (2015). Enhancing emotion-based learning in decision-making under uncertainty. Psicothema, 27(4), 368-373. 

Reviewer 3 Report

Using the theoretical matrix of the Dual Systems Model, the authors present experimental work examining the relationship between decision making and executive functioning in a group of early adolescents and a group of young adults. Overall, the work is interesting and possibly worthy of publication. Before doing so, however, I think it is necessary to clarify the following two points:

The researchers enrolled 140 people in the study, equally divided between first-year secondary school students and first-year university students. It is important to clarify how the sample size required to demonstrate the effects hypothesized in the study was calculated.

To examine executive functions, the authors use the Wisconsin Card Sorting Task (WCST). The choice is justified, but the literature has shown that there are four different executive functions (Di Tella et al. PMID: 33057089; Fisk & Sharp PMID: 15742539). I suggest citing this literature and pointing out that the authors are aware that there are four executive functions and that future work might consider additional tests that measure other executive functions, particularly those related to inhibition.

Author Response

Thank you very much for reviewing our paper, behavsci-2181319, entitled “Risk Decision Making and Executive Function Among Adolescents and Young Adults”, and for giving us the opportunity to submit a revised revision for further consideration in the Behavioral Science. We would like to thank the reviewers for their helpful comments on the manuscript. Below we reproduce all of the reviewers’ comments in bold and place our responses below in regular text.

Using the theoretical matrix of the Dual Systems Model, the authors present experimental work examining the relationship between decision making and executive functioning in a group of early adolescents and a group of young adults. Overall, the work is interesting and possibly worthy of publication. Before doing so, however, I think it is necessary to clarify the following two points:

The researchers enrolled 140 people in the study, equally divided between first-year secondary school students and first-year university students. It is important to clarify how the sample size required to demonstrate the effects hypothesized in the study was calculated.

Thank you for this comment, the sample size was calculated using G*Power 3.1 for a medium effect size, 95 % power, and  p < .05. This is now introduced in the procedure section. (Lines 253-255)

To examine executive functions, the authors use the Wisconsin Card Sorting Task (WCST). The choice is justified, but the literature has shown that there are four different executive functions (Di Tella et al. PMID: 33057089; Fisk & Sharp PMID: 15742539). I suggest citing this literature and pointing out that the authors are aware that there are four executive functions and that future work might consider additional tests that measure other executive functions, particularly those related to inhibition.

Thank you very much for your comment, we added the following text in lines 601-605 and two new references in the references section to point out in future research to address the analysis of subcomponents of executive function that may be associated to risk behavior.

“Furthermore, according to Fisk & Sharp [68], executive functioning can be subdivided into four operations: updating, shifting, inhibition and access.  Previous studies have shown that age is associated to the type of operations, and that the four subcomponents of EFs are associated differently to cognitive abilities, such as the Theory of Mind [69].  In future research, it should be explored which FE operations are associated with risk behavior.” (lines 601-605).

We have added the new references (68 y 69) in the reference section of the manuscript:

Fisk JE, Sharp CA. Age-related impairment in executive functioning: updating, inhibition, shifting, and access. J Clin Exp Neuropsychol. 2004 , 7, 874-90. doi: 10.1080/13803390490510680.

Di Tella M, Ardito RB, Dutto F, Adenzato M. On the (lack of) association between theory of mind and executive functions: a study in a non-clinical adult sample. Sci Rep. 2020 ,10, 17283. doi: 10.1038/s41598-020-74476-0.